# Asymmetric Michael Addition of Malononitrile with Chalcones via Rosin-Derived Bifunctional Squaramide

**Ning Lin \*** , **Qiu-Xiang Wei, Li-Hua Jiang, Yan-Qiu Deng, Zhen-Wei Zhang \*** **and Qing Chen**

College of Pharmacy, Guangxi Zhuang Yao Medicine Center of Engineering and Technology, Guangxi University of Chinese Medicine, Nanning 530200, China; Xiang545800@163.com (Q.-X.W.); lihuajiang12@163.com (L.-H.J.); dengyanqiu0501@163.com (Y.-Q.D.); chenqing@gxtcmu.edu.cn (Q.C.)

\* Correspondence: linning@gxtcmu.edu.cn (N.L.); charliezh@163.com (Z.-W.Z.);
Tel.: +86-188-0771-8996 (N.L.); +86-181-6964-8696 (Z.-W.Z.)

**Abstract:** A rosin-derived bifunctional squaramide catalyzed asymmetric Michael addition of malononitrile with chalcones was discovered. This protocol provides a methodology for the facile synthesis of chiral γ-cyano carbonyl compounds in high yields and enantioselectivities (up to 99% yield and 90% *ee*) with a lower catalyst loading (0.3 mol%). The predominant *R*-configured adducts were obtained by this organocatalystic reaction, according to the experimental findings.

**Keywords:** asymmetric catalysis; michael addition; malononitrile; chalcone; rosin-derived bifunctional catalyst

---

## 1. Introduction

The Michael reaction of carboanion nucleophiles to activated olefins represents a powerful type of the most remarkable transformations for the new carbon–carbon bond formation in modern organic synthesis, and has been immensely exploited over the past few decades [1–7]. Among the versatile nucleophiles, the employment of malononitrile for asymmetric Michael addition has received extensive attention since its nitrile group could be efficiently converted to valuable functionalities [8–27]. To date, a few research groups have devoted their efforts to the catalytic asymmetric Michael reaction of malononitrile onto chalcones and their analogues, by either metal-catalytic [19,20] or organocatalytic [21–27] methods. Despite those gratifying advances, it should be reminded that most of the ligands and organocatalysts utilized in this transformation are commonly cinchona alkaloid-type. Therefore, to seek an efficient catalytic system with a novel organocatalyst is still a challenging and interesting task.

Rosin-derived bifunctional thiourea organocatalysts, originated from the abundantly available natural rosin, were revealed to be highly efficient for some catalytic asymmetric reactions, including Aza-Henry [28], Mannich reaction [29,30], Aldol reaction [31,32], Michael addition [33–36], and Friedel-Crafts alkylation [37]. The thiourea moiety of those organocatalysts is usually introduced at position C-4 of rosin skeleton, while the tertiary amine moiety is either 1,2-diaminocyclohexane 1 or cinchona alkaloid 2 (Figure 1). In addition to thiourea, squaramide is also a good hydrogen-bonding donor and has been successfully applied to facilitate various asymmetric transformations [38–43]. Recently, we have developed several bifunctional squaramide catalysts at position C-4 or C-7 of rosin scaffold, which exhibited excellent enantioselectivities in asymmetric catalytic 1,3-dipolar cycloaddition reactions [44] and Michael/cyclization cascade reactions [45]. However, to the best of our knowledge, rosin-derived chiral squaramides have not been applied for the enantioselective Michael reaction of

malononitrile and chalcones. As our ongoing interest in organocatalysis of rosin-derived catalysts, we herein reported the results from the asymmetric Michael addition of malononitrile with chalcones catalyzed by rosin-derived bifunctional squaramide organocatalysts.

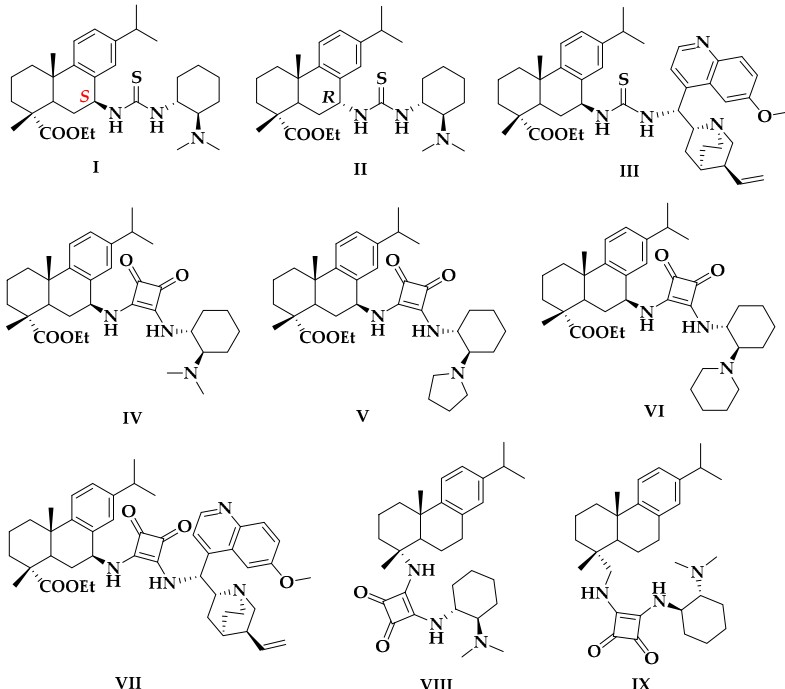

**Figure 1.** Representative thiourea organocatalysts based on rosin skeleton.

## 2. Results and Discussion

### 2.1. Screening of the Catalysts for the Asymmetirc Michael Addtion

Initial investigation started with testing several rosin-derived bifunctional catalysts in a model reaction of malononitrile to trans-chalcone 3a with 10 mol% catalyst loading in $CH_2Cl_2$ at room temperature (Figure 2 and Table 1). The Michael reaction proceeded smoothly by thiourea catalyst I, affording the product 4a with 88% yield and 76% *ee* (entry 1). Unexpectedly, thiourea catalyst II exhibited inferior catalytic activity merely due to the opposite configuration at the position of C-7 of rosin skeleton, and enantioselectivity decreased sharply with the reversed absolute configuration (entry 2, 31% yield, and 26% *ee*). Those results displayed that the configuration at the position of C-7 plays a crucial role in the control of reactivity and enantioselectivity. Thus, we would like to affirm *S*-configuration at C-7 as the optimal for the choice of catalysts.

**Figure 2.** Rosin-derived bifunctional catalysts for this study.

**Table 1.** Effect of catalysts on the model reaction [1].

| Entry | Catalyst | Yield (%) [2] | *ee* (%) [3] | Config. [4] |
|-------|----------|---------------|--------------|-------------|
| 1 | I | 88 | 76 | *R* |
| 2 | II | 31 | 26 | *S* |
| 3 | III | 36.8 | 18 | *R* |
| 4 | IV | >99 | 80 | *R* |
| 5 | V | >99 | 90 | *R* |
| 6 | VI | 85 | 83 | *R* |
| 7 | VII | 20 | 23 | *S* |
| 8 | VIII | 74 | 56 | *R* |
| 9 | IX | 93 | 75 | *R* |

[1] All reactions were conducted with catalyst (10 mol%), malononitrile (0.12 mmol), and trans-chalcone 3a (0.1 mmol) in $CH_2Cl_2$ (2.0 mL) at r.t. for 36 h. No reaction took place without any catalyst. [2] Isolated yields of 4a. [3] Determined by chiral high performance liquid chromatography (HPLC). [4] The absolute configuration was confirmed by HPLC comparisons with the reported data [23].

When thiourea catalyst III was used, both reactivity and enantioselectivity dropped deeply (entry 3 vs. entry 1). This phenomenon may be due to steric effects around the hydrogen-bonding donor of the catalyst derived from more sterically bulky quinine instead of 1,2-diaminocyclohexane. By contrast, squaramide catalysts IV–VI were better catalysts, which could promote this reaction steadily in very good to excellent yields (85%–>99%) and high *ees* (80–90%) (entries 4–6). In terms of reactivity and enantioselectivity, catalyst V gave the best results (entry 5, >99% yield, and 90% *ee*). Notably, like the sterically hindered thiourea catalyst III, squaramide catalyst VII resulted in the same poor outcome of this reaction, but with the reversed absolute configuration (entry 7, 20% yield, and 23% *ee*). Moreover, squaramide catalysts VIII and IX were also surveyed, whose squaramide moiety were introduced at C-4 of rosin skeleton. They could promote the reaction smoothly as well, however, enantioselectivities of the desired product were moderate (entries 8 and 9).

*2.2. Optimization of the Reaction Conditions*

Having identified the optimal squaramide catalyst V for this Michael reaction, we looked forward to subsequent screening of other reaction conditions (Table 2). Optimization studies show that the solvent had a significant influence on the reactivity and enantioselectivity of this transformation (entries 1–7). It was observed that dichloromethane and chloroform were the best solvents. Considering the toxicity of chloroform, dichloromethane was used as the best solvent for further studies. Also, different temperatures were investigated for this reaction, when the reaction was conducted under 0 °C or −20 °C, enantioselectivities would not greatly improve, while the reactivities dramatically declined (entries 8 and 9). Gradually reducing catalyst loading did not make any obvious difference from the enantioselectivities, whereas the yield of the reaction became worse (entries 10–14). To our delight, the catalytic activity could be notably enhanced when the reaction was carried out in 0.5 mL $CH_2Cl_2$ with 0.3 mol% catalyst loading (entry 15). After taking many factors into consideration, including reactivity, enantioselectivity, catalyst loading, and solvent volume, the asymmetric Michael reaction of malononitrile to trans-chalcone 3a could achieve high yield and *ee* value (87% yield and 90% *ee*, entry 15) in the presence of squaramide catalyst V with a lower catalyst loading (0.3 mol%) in $CH_2Cl_2$ (0.5 mL) at room temperature compared with the present literature data [21–27], where only chiral quinine-derived squaramide organocatalyst prepared for the same model reaction by Du [25] could show high activity and enatioselectivity (82% yield and 89% *ee*) under 0.5 mol% catalyst loading.

**Table 2.** Optimization of the reaction conditions [1].

| Entry | x | Solvent | Yield (%) [2] | *ee* (%) [3] |
|---|---|---|---|---|
| 1 | 10 | $CH_2Cl_2$ | 99 | 90 |
| 2 | 10 | $CHCl_3$ | 99 | 90 |
| 3 | 10 | $CH_2ClCH_2Cl$ | 95 | 90 |
| 4 | 10 | MeOH | 88 | 0 |
| 5 | 10 | THF | 32 | 60 |
| 6 | 10 | $Et_2O$ | 21 | 59 |
| 7 | 10 | toluene | 35 | 79 |
| 8 [4] | 10 | $CH_2Cl_2$ | 50 | 91 |
| 9 [5] | 10 | $CH_2Cl_2$ | 15 | 92 |
| 10 | 5 | $CH_2Cl_2$ | 93 | 90 |
| 11 | 1 | $CH_2Cl_2$ | 75 | 90 |
| 12 | 0.5 | $CH_2Cl_2$ | 75 | 90 |
| 13 | 0.3 | $CH_2Cl_2$ | 70 | 90 |
| 14 | 0.1 | $CH_2Cl_2$ | 29 | 90 |
| 15 [6] | 0.3 | $CH_2Cl_2$ | 87 | 90 |

[1] Unless otherwise stated, all reactions were conducted with catalyst (10 mol%), malononitrile (0.12 mmol) and trans-chalcone 3a (0.1 mmol) in $CH_2Cl_2$ (2.0 mL) at r.t. for 36 h. [2] Isolated yields of 4a. [3] Determined by chiral HPLC. [4] 0 °C. [5] −20 °C. [6] Reaction conducted in $CH_2Cl_2$ (0.5 mL).

*2.3. The Scope of the Asymmetric Michael Reaction*

After the optimal reaction conditions were established, the scope of the asymmetric Michael reaction of malononitrile with various trans-chalcones 3a–o and the analogues 3p–q were investigated. The results were summarized in Table 3. The electronic property of different substituents on the aromatic ring ($R^1$) of trans-chalcones 3b–h had no significant effect on the enantioselectivities. In detail, the trans-chalcones bearing electron-donating groups (Me and OMe, entries 2–3) or electron-withdrawing groups (-F, Cl, and Br, entries 4–6) at the 4-position of the aromatic ring led to high enantioselectivities with 85–90% *ee*. However, the radius of the halogen noticeably impacted the reactivities of chalcones. Furthermore, the stereoselectivities and reactivities of this reaction rely on the position of the aryl substituents. The substrates bearing the chloro at 3-position (entry 7), methoxy group at 3-position (entry 8), and 2-position (entry 9) of the aromatic ring were all converted to the corresponding products, but the enantioselectivities and yields markedly decreased due to the position and steric hindrance. When trans-chalcones 3j–n with different electronic substituents on the aromatic ring ($R^2$) were used in this reaction (entries 10–14), the same results were observed as the substituents ($R^1$) in 4-position of the aromatic ring. Substrate 3o with 4-Me substituent on both $R^1$ and $R^2$ phenyl reacted with malononitrile smoothly to afford the corresponding adduct with good yield and high enantioselectivity (70% yield and 86% *ee*, entry 15). Other chalcone analogues, such as 2-enoylpyridine 3p and 1-naphthlaldehyde derivative 3q, were also tested, both giving low results (entries 16–17). The pyridine nitrogen of substrate 3p might involve the hydrogen bonding of the squaramide catalyst V in the H-bond framework to change the steric environment of the catalytic system, which made the activity of catalyst fade out, leading to lower reactivity and enantioselectivity [26]. Poor results of substrate 3q may be due to steric hindrance, similar to substrate 3i.

**Table 3.** Scope of the asymmetric Michael reaction [1].

| Entry | $R^1$ | $R^2$ | Product | Yield (%) [2] | *ee* (%) [3] |
|-------|-------|-------|---------|------------|------------|
| 1 | $C_6H_5$ | $C_6H_5$ | **4a** | 87 | 90 |
| 2 | $4\text{-MeC}_6H_4$ | $C_6H_5$ | **4b** | 97 | 86 |
| 3 | $4\text{-OMeC}_6H_4$ | $C_6H_5$ | **4c** | 92 | 87 |
| 4 | $4\text{-FC}_6H_4$ | $C_6H_5$ | **4d** | 94 | 90 |
| 5 | $4\text{-ClC}_6H_4$ | $C_6H_5$ | **4e** | 72 | 85 |
| 6 | $4\text{-BrC}_6H_4$ | $C_6H_5$ | **4f** | 45 | 85 |
| 7 | $3\text{-ClC}_6H_4$ | $C_6H_5$ | **4g** | 72 | 90 |
| 8 | $3\text{-OMeC}_6H_4$ | $C_6H_5$ | **4h** | 65 | 85 |
| 9 | $2\text{-OMeC}_6H_4$ | $C_6H_5$ | **4i** | 52 | 35 |
| 10 | $C_6H_5$ | $4\text{-MeC}_6H_4$ | **4j** | 64 | 88 |
| 11 | $C_6H_5$ | $4\text{-OMeC}_6H_4$ | **4k** | 76 | 80 |
| 12 | $C_6H_5$ | $4\text{-FC}_6H_4$ | **4l** | 99 | 80 |
| 13 | $C_6H_5$ | $4\text{-ClC}_6H_4$ | **4m** | 57 | 80 |
| 14 | $C_6H_5$ | $4\text{-BrC}_6H_4$ | **4n** | 41 | 79 |
| 15 | $4\text{-MeC}_6H_4$ | $4\text{-MeC}_6H_4$ | **4o** | 70 | 86 |
| 16 | $C_6H_5$ | pyridin-2-yl | **4p** | 55 | 22 |
| 17 | 1-Naphthyl | $C_6H_5$ | **4q** | 50 | 39 |

[1] Unless otherwise stated, all reactions were conducted with catalyst (0.3 mol%), malononitrile (0.12 mmol), and substrates 3 (0.1 mmol) in CH$_2$Cl$_2$ (0.5 mL) at r.t. for 36 h. [2] Isolated yields of 4. [3] Determined by chiral HPLC.

### 2.4. Plausible Transition-State Model of the Asymmetric Michael Reaction

Based on literature reports [25–27,43] and the predominant *R*-configured product 4a, the feasible activation model for the asymmetric Michael reaction was proposed via cooperative catalysis of the squaramide functionality and the tertiary amino group of the rosin-derived squaramide V (Figure 3). The squaramide moiety activated trans-chalcone 3a through bidentate hydrogen bonds. Simultaneously, α-proton of malononitrile was captured by the basic tertiary nitrogen to form an active carbanion. Subsequent addition of the carbanion to the Si-face of 3a led to the desired adduct as major stereoisomer, which is in agreement with the observed experimental results.

**Figure 3.** Plausible transition-state model.

## 3. Experimental Section

### 3.1. General Information

Unless noted otherwise, all commercial reagents were purchased from chemical reagent suppliers (Alfa Aesar Chemical Co. Ltd., Shanghai, China; Sigma-Aldrich Chemical Company, Darmstadt, Germany; and Aladdin Chemical Co. Ltd., Shanghai, China) and used as received, without further purification. Isolation of the crude products was accomplished by flash chromatography on silica gel (200–300 mesh, Qingdao Sea Chemical Reagent Co. Ltd., Qingdao, Shandong, China). Thin layer chromatography (TLC) analysis was carried out using EM separations percolated TLC sheets (silica gel GF254, 0.2 mm, Qingdao Sea Chemical Reagent Co. Ltd., Qingdao, Shandong, China) under UV light. $^1$H NMR spectra were obtained with a Bruker Avance III spectrometer (400 MHz, Switzerland). Chemical shifts were published as parts per million (ppm) in $\delta$ units internally, with tetramethylsilane (TMS, $\delta$ = 0.00 ppm) as the referenced standard. The enantiomeric excesses (*ee*) were determined by HPLC analyses using a Shimadzu 10A instrument (Japan) with Daicel Chiralcel OD-H or AD-H column (0.46 cm diameter × 25 cm length) in comparison with racemic samples and *n*-hexane/*i*-PrOH as the eluent. Known adducts 4b, 4e, and 4j were assigned as *R*-configuration by HPLC comparisons with the reported data [23], respectively, and the absolute configurations of other products were confirmed by analogy with compounds 4b, 4e, and 4j. All the rosin-derived chiral squaramide organocatalysts I–IX were synthesized according to the literature procedures [25,28,29,46]. All $^1$H NMR and HPLC spectra of compounds 4a-q could be found in Supplementary Materials.

### 3.2. Typical Procedure for the Michael Addition

Catalyst V (7.1 mg, 0.012 mmol) was dissolved into dicloromethane to prepare the solution of catalyst V (20.0 mL, 0.6 mmol/L). To the above solution (0.5 mL, containing catalyst 0.0003 mmol, 0.3 mol%) was added malononitrile (8 mg, 0.12 mmol), chalcones 3 (0.1 mmol) subsequently. The resulting mixture was then stirred at room temperature for 36 h. The corresponding adducts 4 were isolated through flash silica gel chromatography (eluent, ethyl acetate/petroleum ether).

*(R)-2-(3-oxo-1,3-diphenylpropyl)malononitrile* (4a) [19,25]: 99% yield; white solid; 90% *ee*, determined by HPLC (Daicel Chiralcel OD-H, *n*-hexane/*i*-PrOH = 80/20, 25 °C, 0.8 mL min$^{-1}$, 254 nm): $t_R$ = 23.8 min (*major*), 38.6 min (*minor*). $^1$H NMR (400 MHz, CDCl$_3$): $\delta$ 7.97 (d, *J* = 7.3 Hz, 2H), 7.65–7.60 (m, 1H), 7.52–7.42 (m, 7H), 4.65 (d, *J* = 5.1 Hz, 1H), 3.96 (dt, *J* = 8.1, 5.2 Hz, 1H), 3.71–3.66 (m, 2H).

*(R)-2-(3-oxo-3-phenyl-1-p-tolylpropyl)malononitrile* (4b) [19,23,25]: 97% yield; colorless oil; 86% *ee*, determined by HPLC (Daicel Chiralcel AD-H, *n*-hexane/*i*-PrOH = 80/20, 25 °C, 0.8 mL min$^{-1}$, 254 nm): $t_R$ = 10.8 min (*major*), 15.6 min (*minor*). $^1$H NMR (400 MHz, CDCl$_3$): $\delta$ 7.98–7.93 (m, 2H), 7.64–7.58 (m, 1H), 7.48 (dd, *J* = 10.6, 4.8 Hz, 2H), 7.33 (d, *J* = 8.1 Hz, 2H), 7.24–7.21 (m, 2H), 4.62–4.56 (m, 1H), 3.92 (dt, *J* = 8.2, 5.4 Hz, 1H), 3.67–3.52 (m, 2H), 2.35 (s, 3H).

*(R)-2-(1-(4-methoxyphenyl)-3-oxo-3-phenylpropyl)malononitrile* (4c) [19,25]: 92% yield; colorless oil; 86% *ee*, determined by HPLC (Daicel Chiralcel AD-H, *n*-hexane/*i*-PrOH = 80/20, 25 °C, 0.8 mL min$^{-1}$, 254 nm): $t_R$ = 14.2 min (*major*), 23.2 min (*minor*). $^1$H NMR (400 MHz, CDCl$_3$): $\delta$ 8.02–7.93 (m, 2H), 7.66–7.59 (m, 1H), 7.50 (dd, *J* = 10.6, 4.8 Hz, 2H), 7.40–7.34 (m, 2H), 6.99–6.91 (m, 2H), 4.61 (d, *J* = 5.0 Hz, 1H), 3.92 (dt, *J* = 8.5, 5.2 Hz, 1H), 3.82 (s, 3H), 3.67–3.63 (m, 2H).

*(R)-2-(1-(4-fluorophenyl)-3-oxo-3-phenylpropyl)malononitrile* (4d) [19,25]: 94% yield; white solid; 90% *ee*, determined by HPLC (Daicel Chiralcel AD-H, *n*-hexane/*i*-PrOH = 80/20, 25 °C, 0.8 mL min$^{-1}$, 254 nm): $t_R$ = 10.4 min (*major*), 16.4 min (*minor*). $^1$H NMR (400 MHz, CDCl$_3$): $\delta$ 7.97 (dd, *J* = 8.3, 1.1 Hz, 2H), 7.66–7.60 (m, 1H), 7.50 (t, *J* = 7.7 Hz, 2H), 7.47–7.42 (m, 2H), 7.15–7.09 (m, 2H), 4.62 (d, *J* = 5.1 Hz, 1H), 4.01–3.93 (m, 1H), 3.72–3.60 (m, 2H).

*(R)-2-(1-(4-chlorophenyl)-3-oxo-3-phenylpropyl)malononitrile* (4e) [19,23,25]: 72% yield; white solid; 85% *ee*, determined by HPLC (Daicel Chiralcel AD-H, *n*-hexane/*i*-PrOH = 80/20, 25 °C, 0.8 mL min$^{-1}$, 254 nm):

$t_R$ = 11.4 min (*major*), 18.7 min (*minor*). [1]H NMR (400 MHz, CDCl$_3$): δ 7.96 (dd, *J* = 8.3, 1.2 Hz, 2H), 7.64 (s, 1H), 7.51 (t, *J* = 7.7 Hz, 2H), 7.41 (d, *J* = 1.5 Hz, 4H), 4.63 (d, *J* = 5.1 Hz, 1H), 3.95 (d, *J* = 8.4 Hz, 1H), 3.68–3.63 (m, 2H).

*(R)-2-(1-(4-bromophenyl)-3-oxo-3-phenylpropyl)malononitrile* (4f) [25]: 45% yield; white solid; 85% *ee*, determined by HPLC (Daicel Chiralcel AD-H, *n*-hexane/*i*-PrOH = 80/20, 25 °C, 0.8 mL min$^{-1}$, 254 nm): $t_R$ = 11.6 min (*major*), 18.8 min (*minor*). [1]H NMR (400 MHz, CDCl$_3$): δ 7.96 (d, *J* = 7.9 Hz, 2H), 7.64 (t, *J* = 7.2 Hz, 1H), 7.57 (d, *J* = 8.2 Hz, 2H), 7.50 (t, *J* = 7.6 Hz, 2H), 7.34 (d, *J* = 8.2 Hz, 2H), 4.62 (d, *J* = 5.0 Hz, 1H), 3.93 (dd, *J* = 8.3, 5.2 Hz, 1H), 3.67–3.59 (m, 2H).

*(R)-2-(1-(3-chlorophenyl)-3-oxo-3-phenylpropyl)malononitrile* (4g) [19]: 72% yield; white solid; 90% *ee*, determined by HPLC (Daicel Chiralcel AD-H, *n*-hexane/*i*-PrOH = 80/20, 25 °C, 0.8 mL min$^{-1}$, 254 nm): $t_R$ = 9.8 min (*major*), 12.2 min (*minor*). [1]H NMR (400 MHz, CDCl$_3$): δ 7.95 (dd, *J* = 8.3, 1.1 Hz, 2H), 7.60 (d, *J* = 7.4 Hz, 1H), 7.50–7.43 (m, 3H), 7.38–7.33 (m, 3H), 4.61 (d, *J* = 5.2 Hz, 1H), 3.98–3.90 (m, 1H), 3.66–3.62 (m, 2H).

*(R)-2-(1-(3-methoxyphenyl)-3-oxo-3-phenylpropyl)malononitrile* (4h) [19]: 65% yield; white solid; 85% *ee*, determined by HPLC (Daicel Chiralcel AD-H, *n*-hexane/*i*-PrOH = 90/10, 25 °C, 1.0 mL min$^{-1}$, 254 nm): $t_R$ = 16.9 min (*major*), 19.5 min (*minor*). [1]H NMR (400 MHz, CDCl$_3$): δ 7.98–7.93 (m, 2H), 7.63–7.58 (m, 1H), 7.48 (dd, *J* = 10.7, 4.8 Hz, 2H), 7.33 (t, *J* = 8.0 Hz, 1H), 7.03–6.89 (m, 3H), 4.62 (d, *J* = 5.2 Hz, 1H), 3.95–3.89 (m, 1H), 3.82 (s, 3H), 3.66–3.64 (m, 2H).

*(R)-2-(1-(2-methoxyphenyl)-3-oxo-3-phenylpropyl)malononitrile* (4i) [19,25]: 52% yield; white solid; 35% *ee*, determined by HPLC (Daicel Chiralcel AD-H, *n*-hexane/*i*-PrOH = 90/10, 25 °C, 1.0 mL min$^{-1}$, 254 nm): $t_R$ = 11.7 min (*major*), 13.2 min (*minor*). [1]H NMR (400 MHz, CDCl$_3$): δ 7.97–7.92 (m, 2H), 7.58 (d, *J* = 7.4 Hz, 1H), 7.47 (t, *J* = 7.7 Hz, 2H), 7.35–7.29 (m, 2H), 6.99–6.91 (m, 2H), 4.66 (d, *J* = 6.6 Hz, 1H), 4.44 (d, *J* = 6.8 Hz, 1H), 3.88 (s, 3H), 3.73–3.64 (m, 2H).

*(R)-2-(3-oxo-1-phenyl-3-p-tolylpropyl)malononitrile* (4j) [47]: 64% yield; colorless oil; 88% *ee*, determined by HPLC (Daicel Chiralcel AD-H, *n*-hexane/*i*-PrOH = 80/20, 25 °C, 0.8 mL min$^{-1}$, 254 nm): $t_R$ = 12.3 min (*major*), 18.5 min (*minor*). [1]H NMR (400 MHz, CDCl$_3$): δ 7.86 (d, *J* = 8.2 Hz, 2H), 7.46–7.38 (m, 5H), 7.28 (d, *J* = 8.0 Hz, 2H), 4.65 (d, *J* = 5.1 Hz, 1H), 3.97–3.88 (m, 1H), 3.66–3.61 (m, 2H), 2.42 (s, 3H).

*(R)-2-(3-(4-methoxyphenyl)-3-oxo-1-phenylpropyl)malononitrile* (4k) [25]: 76% yield; colorless oil; 80% *ee*, determined by HPLC (Daicel Chiralcel AD-H, *n*-hexane/*i*-PrOH = 80/20, 25 °C, 0.8 mL min$^{-1}$, 254 nm): $t_R$ = 19.9 min (*major*), 31.5 min (*minor*). [1]H NMR (400 MHz, CDCl$_3$): δ 7.95 (d, *J* = 9.0 Hz, 2H), 7.44 (d, *J* = 5.8 Hz, 5H), 6.95 (d, *J* = 8.9 Hz, 2H), 4.69 (d, *J* = 5.0 Hz, 1H), 3.97–3.91 (m, 1H), 3.88 (s, 3H), 3.65–3.59 (m, 2H).

*(R)-2-(3-(4-fluorophenyl)-3-oxo-1-phenylpropyl)malononitrile* (4l) [25]: 99% yield; colorless oil; 80% *ee*, determined by HPLC (Daicel Chiralcel AD-H, *n*-hexane/*i*-PrOH = 80/20, 25 °C, 0.8 mL min$^{-1}$, 254 nm): $t_R$ = 11.5 min (*major*), 13.6 min (*minor*). [1]H NMR (400 MHz, CDCl$_3$): δ 8.04–7.96 (m, 2H), 7.47–7.38 (m, 5H), 7.20–7.13 (m, 2H), 4.62 (d, *J* = 5.2 Hz, 1H), 3.99–3.92 (m, 1H), 3.67–3.63 (m, 2H).

*(R)-2-(3-(4-chlorophenyl)-3-oxo-1-phenylpropyl)malononitrile* (4m) [25]: 57% yield; white solid; 80% *ee*, determined by HPLC (Daicel Chiralcel AD-H, *n*-hexane/*i*-PrOH = 80/20, 25 °C, 0.8 mL min$^{-1}$, 254 nm): $t_R$ = 12.9 min (*major*), 15.5 min (*minor*). [1]H NMR (400 MHz, CDCl$_3$): δ 7.92–7.87 (m, 2H), 7.48–7.44 (m, 2H), 7.44–7.39 (m, 5H), 4.60 (d, *J* = 5.2 Hz, 1H), 3.98–3.91 (m, 1H), 3.65–3.62 (m, 2H).

*(R)-2-(3-(4-bromophenyl)-3-oxo-1-phenylpropyl)malononitrile* (4n) [19,25]: 41% yield; white solid; 79% *ee*, determined by HPLC (Daicel Chiralcel AD-H, *n*-hexane/*i*-PrOH = 80/20, 25 °C, 0.8 mL min$^{-1}$, 254 nm): $t_R$ = 14.3 min (*major*), 17.2 min (*minor*). [1]H NMR (400 MHz, CDCl$_3$): δ 7.80 (d, *J* = 8.4 Hz, 2H), 7.61 (d, *J* = 8.4 Hz, 2H), 7.41 (s, 5H), 4.58 (d, *J* = 5.2 Hz, 1H), 3.97–3.89 (m, 1H), 3.63–3.60 (m, 2H).

*(R)-2-(3-oxo-1,3-di-p-tolylpropyl)malononitrile* (4o) [25]: 70% yield; white solid; 86% *ee*, determined by HPLC (Daicel Chiralcel AD-H, *n*-hexane/*i*-PrOH = 90/10, 25 °C, 1.0 mL min$^{-1}$, 254 nm): $t_R$ = 14.6 min

(*major*), 22.3 min (*minor*). [1]H NMR (400 MHz, CDCl$_3$): δ 7.84 (d, *J* = 8.2 Hz, 2H), 7.32 (d, *J* = 8.1 Hz, 2H), 7.26 (d, *J* = 8.0 Hz, 2H), 7.20 (d, *J* = 7.9 Hz, 2H), 4.59 (d, *J* = 5.1 Hz, 1H), 3.89 (d, *J* = 8.3 Hz, 1H), 3.62–3.58 (m, 2H), 2.40 (s, 3H), 2.34 (s, 3H).

*(R)-2-(3-oxo-1-phenyl-3-(pyridin-2-yl)propyl)malononitrile* (4p) [26]: 55% yield; white solid; 22% *ee*, determined by HPLC (Daicel Chiralcel AD-H, *n*-hexane/*i*-PrOH = 90/10, 25 °C, 1.0 mL min$^{-1}$, 254 nm): t$_R$ = 17.9 min (*major*), 20.8 min (*minor*). [1]H NMR (400 MHz, CDCl$_3$): δ 8.69 (ddd, *J* = 4.7, 1.6, 0.9 Hz, 1H), 8.03–7.99 (m, 1H), 7.85 (td, *J* = 7.7, 1.7 Hz, 1H), 7.52 (ddd, *J* = 7.6, 4.8, 1.2 Hz, 1H), 7.48–7.45 (m, 2H), 7.44–7.37 (m, 3H), 4.52 (d, *J* = 5.1 Hz, 1H), 4.08 (dd, *J* = 17.5, 4.7 Hz, 1H), 3.97–3.87 (m, 2H).

*(R)-2-(1-(naphthalen-1-yl)-3-oxo-3-phenylpropyl)malononitrile* (4q) [25]: 50% yield; white solid; 39% *ee*, determined by HPLC (Daicel Chiralcel AD-H, *n*-hexane/*i*-PrOH = 90/10, 25 °C, 1.0 mL min$^{-1}$, 254 nm): t$_R$ = 15.4 min (*major*), 17.4 min (*minor*). [1]H NMR (400 MHz, CDCl$_3$): δ 8.13 (d, *J* = 8.5 Hz, 1H), 8.00–7.96 (m, 2H), 7.91 (dd, *J* = 18.2, 8.0 Hz, 2H), 7.70 (d, *J* = 7.1 Hz, 1H), 7.66–7.56 (m, 3H), 7.50 (dt, *J* = 10.1, 7.8 Hz, 3H), 5.03 (d, *J* = 6.7 Hz, 1H), 4.70 (d, *J* = 5.3 Hz, 1H), 3.86–3.83 (m, 2H).

## 4. Conclusions

In summary, we have developed an effective asymmetric Michael addition of malononitrile onto various trans-chalcones and their analogues catalyzed by chiral squaramide derived from commercially available rosin under mild conditions with a low catalyst loading (0.3 mol%), affording optical products (up to 90% *ee*). Further studies on bifunctional rosin-derived chiral squaramide organocatalysts for other enantioselective reactions are currently underway in our laboratory.

**Supplementary Materials:** The following are available online at http://www.mdpi.com/2073-4344/10/1/14/s1, containing NMR spectra of compounds 4a–q, and HPLC spectra of racemic and chiral products 4a–q.

**Author Contributions:** Conceptualization, N.L. and Z.-W.Z.; methodology, N.L., Q.-X.W., and L.-H.J.; validation, Q.-X.W. and L.-H.J.; formal analysis, Q.-X.W., L.-H.J., and Y.-Q.D.; investigation, N.L., Q.-X.W., and L.-H.J.; resources, N.L. and Q.C.; data curation, Q.-X.W. and L.-H.J.; writing—original draft preparation, Z.-W.Z.; writing—review and editing, N.L.; visualization, Q.C.; supervision, N.L.; project administration, N.L.; funding acquisition, N.L., Z.-W.Z., and Q.C. All authors have read and agreed to the published version of the manuscript.

**Funding:** This research was funded by National Natural Science Foundation of China, grant number 21861009; Guangxi Natural Science Foundation, grant number 2018GXNSFAA281317 and 2018GXNSFBA138032; Guangxi University of Chinese Medicine Research Foundation for introduced Ph.D., grant number XB0170027; Innovation Project of Guangxi Graduate Education, grant number YCSW2019175; Guangxi University of Chinese Medicine First-class Discipline Construction of Chinese Medicine.

**Conflicts of Interest:** The authors declare no conflict of interest.

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
