# Peer review of "Asymmetric Michael Addition of Malononitrile with Chalcones via Rosin-Derived Bifunctional Squaramide"

_catalysts, doi:10.3390/catal10010014_

Round 1

Reviewer 1 Report

The manuscript by Ning Lin et al. describes the screening of several rosin-derived organocatalysts with thiourea or squaramide moieties. The research is well designed and the results are clearly presented. Some point of consideration are:

It is not clear from the manuscript if the catalysts I-IX were prepared by the authors using their method or the literature procedure. In any case, references of the procedures themselves should be provided in the experimental part. The authors should provide the results of a blank experiment without any catalyst in Table 1. In the introduction section, the authors mention that the catalytic asymmetric Michael reaction of malononitrile and chalcones was already studied with other organocatalysts [21-27]. In the discussion section of the manuscript, some comparison of the proposed catalyst and literature data in terms of activity, enantioselectivity, yields, etc. would be quite informative.

Author Response

The manuscript by Ning Lin et al. describes the screening of several rosin-derived organocatalysts with thiourea or squaramide moieties. The research is well designed and the results are clearly presented. Some point of consideration are:

 1. It is not clear from the manuscript if the catalysts I-IX were prepared by the authors using their method or the literature procedure. In any case, references of the procedures themselves should be provided in the experimental part.

Answer: Thanks for the valuable comments. We have added the reference information (ref. 25, 28-29, 46) of the relevant synthesis procedures of the organocatalysts I-IX in the experimental part (Line 154-155). 

2. The authors should provide the results of a blank experiment without any catalyst in Table 1.

Answer: We thank the reviewer for his/her valuable advice. In our experiments, this reaction did not proceed without any catalyst. We have added the description in Table 1 (Line 74). 

3. In the introduction section, the authors mention that the catalytic asymmetric Michael reaction of malononitrile and chalcones was already studied with other organocatalysts [21-27]. In the discussion section of the manuscript, some comparison of the proposed catalyst and literature data in terms of activity, enantioselectivity, yields, etc. would be quite informative.

Answer: Thanks for the valuable comments. We have added the information of comparison of our catalytic system with the relevant literatures in the discussion section (Line 88-95).

Reviewer 2 Report

This manuscript reports on the evaluation of an intersting group of novel rosin-based organocatalysts promoting enantioselective conjugate addition of malonitrile to chalchones. By means of sequential optimisations the authors disclosed characteristic structure-activity correlations. On the basis of these valuable results this well-done manuscript is worth for publication, however prior to final acceptance the following points need to be addressed: (i) a brief comment should be inserted at least as an attempted interpretation of the substantial decrease in the "ee" value measured for the formation of the pyridin-2-yl product 4p as presented in Table 3 (entry 16) (e.g. by the involvement of the pyridine nitrogen in the H-bond framework); (ii) the article published in RSC ADVANCES 5 : 115 pp. 95079-95086. , 8 p. (2015) should be added to the citations [38-42] with its extension to [38-43]; (iii) Figure 3 need to be completed by the insertion of a double bond in the four-membered ring.

Author Response

This manuscript reports on the evaluation of an intersting group of novel rosin-based organocatalysts promoting enantioselective conjugate addition of malonitrile to chalchones. By means of sequential optimisations the authors disclosed characteristic structure-activity correlations. On the basis of these valuable results this well-done manuscript is worth for publication, however prior to final acceptance the following points need to be addressed:

 (i) a brief comment should be inserted at least as an attempted interpretation of the substantial decrease in the "ee" value measured for the formation of the pyridin-2-yl product 4p as presented in Table 3 (entry 16) (e.g. by the involvement of the pyridine nitrogen in the H-bond framework).

Answer: We thank the reviewer for his/her valuable comments. We have already mentioned the possible cause of the decrease in the "ee" value of the pyridin-2-yl product 4p according to ref. 26, and we have made some modification according to the reviewer’s advice (Line 119-122). 

(ii) the article published in RSC ADVANCES 5 : 115 pp. 95079-95086. , 8 p. (2015) should be added to the citations [38-42] with its extension to [38-43].

Answer: We thank the reviewer for his/her valuable comments. The relevant literature on mechanistic investigations of a bifunctional squaramide organocatalyst in asymmetric Michael reaction (ref. 43) has been added in the manuscript (Line 39, and 368-370). 

(iii) Figure 3 need to be completed by the insertion of a double bond in the four-membered ring.

Answer: We thank the reviewer for his/her valuable reminding. We have amended in the manuscript (Line 138-139).

Reviewer 3 Report

The Authors report the use of rosin-derived squaramide compounds as catalysts in Michael additions of malononitrile to calchones. Though the obtained enantiomeric excesses (up to 90%) do not represent any improvement with respect to similar catalysts described in literature, the paper gives a good picture on the usefulness of these rosin-derived catalysts.

My only concern regards the poor quality of the NMR spectra reported in the Supplementary Materials and their description in the Experimental Section.  No enlargements of the significant signals are provided to allow the reviewer to check the signal multiplicity; for example, I have some doubts on the description of the signal of the two methylene hydrogens adjacent to the carbonyl group which resonate at about 3.60-3.70 ppm. They are described as qd in 4a, dd in 4e, dd in 4g, dd in 4h, two signals both d in 4i, dd in 4j, dd in 4k, t in 4l, dd in 4m, dd in 4n, dd in 4o. The coupling constants vary significantly from one compound to another, although I would have expected very similar values. These two hydrogens are the AB part of an ABMX system and should show the corresponding classical pattern with a germinal and two vicinal coupling constants.   

Overall, I encourage to resolve the NMR issue and recommend this manuscript for publication on Catalysts.

Author Response

Answer: We thank the reviewer for his/her valuable comments. The two methylene hydrogens adjacent to the carbonyl group belong to the AB part of an ABMX system and often show the corresponding classical pattern with a germinal and two vicinal coupling constants as the reviewer have mentioned. And their 1H NMR signal is often interlaced. After careful check of our NMR data and comparison with related literature (ref. 19, 23, 25), the signals of these two hydrogens which resonate at about 3.60-3.70 ppm could be marked as multiple peaks.

Round 2

Reviewer 1 Report

The authors have taken care of the comments raised by the reviewer and the manuscript can now be accepted.